# The mammalian lectin galectin-8 induces RANKL expression, osteoclastogenesis, and bone mass reduction in mice

Yaron Vinik[1], Hadas Shatz-Azoulay[1], Alessia Vivanti[1], Navit Hever[1], Yifat Levy[1], Rotem Karmona[1], Vlad Brumfeld[2], Saja Baraghithy[3], Malka Attar-Lamdar[3], Sigalit Boura-Halfon[1], Itai Bab[3], Yehiel Zick[1]*

[1]Department of Molecular Cell Biology, Weizmann Institute of Science, Rehovot, Israel; [2]Department of Chemical Research Support, Weizmann Institute of Science, Rehovot, Israel; [3]Bone Laboratory, The Hebrew University of Jerusalem, Jerusalem, Israel

**Abstract** Skeletal integrity is maintained by the co-ordinated activity of osteoblasts, the bone-forming cells, and osteoclasts, the bone-resorbing cells. In this study, we show that mice overexpressing galectin-8, a secreted mammalian lectin of the galectins family, exhibit accelerated osteoclasts activity and bone turnover, which culminates in reduced bone mass, similar to cases of postmenopausal osteoporosis and cancerous osteolysis. This phenotype can be attributed to a direct action of galectin-8 on primary cultures of osteoblasts that secrete the osteoclastogenic factor RANKL upon binding of galectin-8. This results in enhanced differentiation into osteoclasts of the bone marrow cells co-cultured with galectin-8-treated osteoblasts. Secretion of RANKL by galectin-8-treated osteoblasts can be attributed to binding of galectin-8 to receptor complexes that positively (uPAR and MRC2) and negatively (LRP1) regulate galectin-8 function. Our findings identify galectins as new players in osteoclastogenesis and bone remodeling, and highlight a potential regulation of bone mass by animal lectins.

*For correspondence: yehiel. zick@weizmann.ac.il

**Competing interests:** The authors declare that no competing interests exist.

**Reviewing editor**: Gordana Vunjak-Novakovic, Columbia University, United States

## Introduction

Bone is a dynamic tissue that constantly undergoes remodeling by osteoclast-mediated bone resorption and osteoblast-mediated bone formation (*Eriksen, 2010*; *Nakahama, 2010*; *Raggatt and Partridge, 2010*). In a rapidly growing and mature mammals, bone remodeling is positive or balanced, respectively, allowing for bone mass accrual and later for its maintenance. Negatively balanced bone remodeling is a hallmark of pathologies such as osteoporosis and cancerous osteolysis (*Kozlow and Guise, 2005*; *Novack and Teitelbaum, 2008*; *Sturge et al., 2011*).

Skeletal tissues are composed largely of extracellular matrix (ECM). Fibrillar ECM proteins, predominantly type I collagen in bone and type II collagen in cartilage, provide structural integrity and account for mechanical strength. The ECM of bone also contains matricellular proteins that primarily serve as biological modulators. Matricellular proteins interact with cell-surface receptors, such as integrins, the structural matrix, and soluble extracellular factors including growth factors and proteases (*Bornstein and Sage, 2002*). Through these multiple interactions, matricellular proteins modulate cell function and regulate the availability and activity of proteins sequestered in the matrix. Therefore, matricellular proteins contribute to skeletal development, homeostasis, and fracture healing (*Alford and Hankenson, 2006*).

Galectins are a family of glycan-binding proteins secreted by a variety of cell types (*Boscher et al., 2011*; *Di Lella et al., 2011*). As such, they can act as biological cross-linkers for ECM proteins

**eLife digest** The forces applied to the body during daily activities cause bones to be constantly remodeled, which is essential for keeping them healthy. In most adult organisms, new bone is created at the same rate at which old bone is destroyed. This means that overall bone mass remains the same. But, in diseases such as osteoporosis or bone cancer, bone is destroyed more rapidly than at which new bone is made. This leads to brittle bones that are more likely to fracture. Understanding how to increase the rate of bone renewal might therefore help scientists develop new treatments for bone diseases.

Bone is created by cells called osteoblasts and destroyed by other cells called osteoclasts. Both of these types of cells develop from stem cells in the bone marrow. The activity of these cells is controlled by a number of factors, including the matrix of proteins that holds bone together. A group of proteins called galectins are known to act as a bridge between some of the matrix proteins and molecules on the surface of the cells.

Vinik et al. took osteoblasts from a mouse skull, grew them in the laboratory, and then exposed them to a galectin protein called galectin-8. This made the osteoblasts release a protein called RANKL, which is known to boost osteoclast activity. When osteoblasts that had been exposed to galectin-8 were grown alongside bone marrow stem cells, more of the stem cells developed into the bone-destroying osteoclasts.

Mice that were genetically engineered to produce more galectin-8 than normal mice develop brittle bones, despite also creating new bone at a higher rate than do normal mice. This is because osteoclast activity increases at a greater rate, resulting in an overall loss of bone in these animals. This is similar to what occurs in some individuals with osteoporosis. These experiments therefore suggest that galectin-8 plays an important role in bone remodeling and that it may be a potential target for drugs that treat diseases that weaken bones.

and cell-surface receptors (*Elola et al., 2007*). Indeed, certain galectins were reported to function as matricellular proteins (*Troncoso et al., 2014*). The minimal structures recognized by these lectins are β-galactosides displayed on the cell surface as part of more complex glycoconjugates (*Di Lella et al., 2011*). The prototype galectins (galectin-1, -2, -5, -7, -10, -11, -13, -14, and -15) exist as monomers or homodimers of one carbohydrate recognition domain (CRD). The tandem-repeat-type galectins (galectin-4, -6, -8, -9, and -12) harbor two distinct CRDs joined by a peptide linker. The chimera-type galectin-3 consists of a CRD connected to a polypeptide that can pentamerize upon binding to glycan ligands (*Rabinovich and Vidal, 2011*; *Thiemann and Baum, 2011*; *Ledeen et al., 2012*).

Being secreted proteins, as well as proteins having intracellular roles, galectins affect a wide range of biological functions including regulation of cell adhesion, migration, cell growth, apoptosis, and autophagy (*Rabinovich and Vidal, 2011*; *Thiemann and Baum, 2011*; *Thurston et al., 2012*). Still, regulation of bone physiology by galectins has been addressed only to a limited extent. Galectin-9 has been shown to induce osteoblast differentiation initiated by coupling of CD44 to bone morphogenetic protein (BMP) receptors (*Tanikawa et al., 2010*), whereas GC-1 and GC-8, the chicken orthologs of galectin-1 and galectin-8, respectively, were shown to mediate the formation and patterning of pre-cartilage mesenchymal condensations in the developing limb of chicken (*Bhat et al., 2011*).

To study the possible role of galectins as regulators of bone physiology, we focused upon galectin-8 (gal-8), initially cloned in our laboratory, which is a tandem-repeat-type galectin having two sugar-binding domains joined by a linker peptide (*Hadari et al., 1995*; *Levy et al., 2006*). Upon secretion, galectin-8 is equipotent to fibronectin in promoting cell adhesion by ligation and clustering of a selective subset of cell-surface integrins and ECM proteins (*Hadari et al., 2000*; *Levy et al., 2001*; *Eshkar Sebban et al., 2007*). Complex formation between galectin-8 and integrins triggers integrin-mediated signaling cascades (*Levy et al., 2001*, *2003*) that affect cell growth, receptor trafficking, and metastatic potential (*Boura-Halfon et al., 2003*; *Zick et al., 2004*; *Arbel-Goren et al., 2005*; *Reticker-Flynn et al., 2012*).

In this study, we show that galectin-8 regulates bone mass by inducing the secretion of the osteoclastogenic factor, receptor activator of NF-κB ligand (RANKL) (*Hanada et al., 2010*), from

isolated osteoblasts in a cell autonomous manner. As a result, co-culture of galectin-8-treated osteoblasts with bone marrow cells increases their differentiation into active osteoclasts. These effects involve the binding of galectin-8 to the osteoblasts' urokinase plasminogen-activated receptor (uPAR); mannose receptor C, type 2 (MRC2); and the low-density lipoprotein receptor-related protein 1 (LRP1), and seem to be of physiological relevance because galectin-8 transgenic animals exhibit increased expression of RANKL, increased osteoclastogenic activity, and enhanced bone turnover that culminates in reduced bone mass. These data identify galectin-8 as a potential drug target for the prevention of diseases associated with excessive bone loss.

## Results

### Effects of galectin-8 on cultured osteoblasts

To study the effects of galectin-8 on osteoblasts in culture, osteoblasts derived from calvaria of newborn CD1 mice were incubated with 50 nM galectin-8. As shown in *Figure 1A*, such treatment increased by sixfold the expression of RANKL in these cells by 4 hr and resulted in a 2.5-fold increase in secretion of soluble RANKL into the medium (*Figure 1B*) by 24 hr. Extended incubation with galectin-8, up to 6 days maintained the high levels of expression of RANKL (*Figure 1C*). Galectin-8

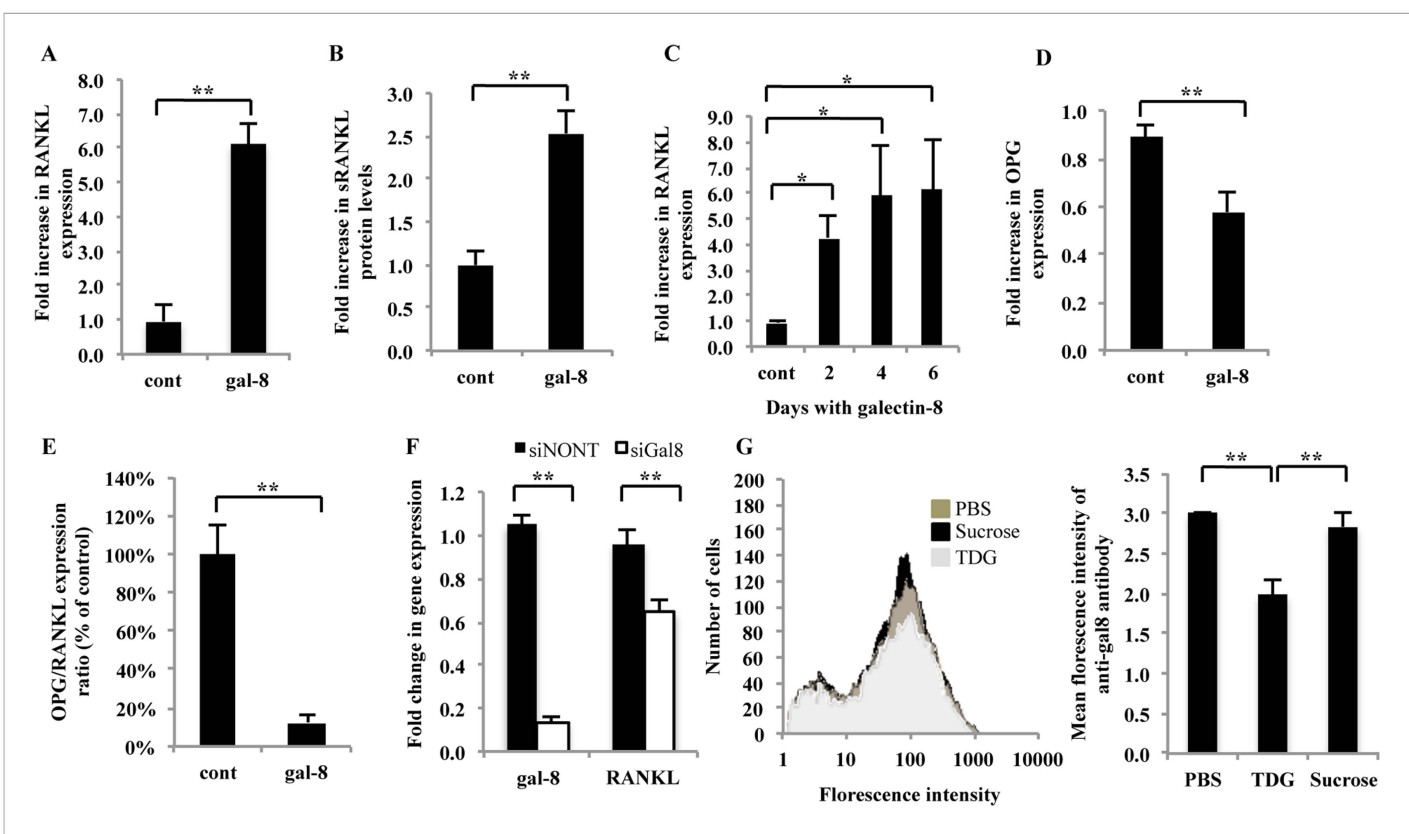

**Figure 1**. Effects of galectin-8 on RANKL and OPG expression in osteoblasts. Osteoblasts derived from calvaria of newborn mice were treated with 50 nM of galectin-8 for 4 hr (**A**, **D**); 24 hr (**B**); or for the indicated times (**C**). After treatments, RNA was extracted and qRT-PCR was conducted in order to quantify changes in expression of RANKL (**A**, **C**) or osteoprotegerin (OPG) (**D**). Actin served as a control for normalization purposes. The levels of soluble RANKL in the medium were quantified by ELISA (**B**). (**E**) OPG/RANKL expression ratio was calculated from the results of **A** and **D**. (**F**) Osteoblasts from the calvaria of newborn mice were grown in 12-well plates (5 × 10⁴ cells per well). After 24 hr, cells were transfected with siRNA for galectin-8. Non-targeting siRNA (siNONT) served as the control. 96 hr thereafter, cells were harvested, RNA was extracted, and qRT-PCR was conducted to quantify changes in mRNA levels of galectin-8 and RANKL. The content of actin mRNA served as a control for normalization purposes. (**G**) Bone marrow cells were extracted and analyzed by flow cytometry for the surface expression of galectin-8. Cells were treated with TDG or sucrose (10 mM in PBS) or just PBS before fixation. Results shown are of a representative histogram of cell number vs florescence intensity of the secondary antibody (left) and quantitation of the averages florescence intensity of the secondary antibody in two experiments carried out in duplicates (right). Results are mean values ± SEM of five (**F**) or six independent experiments (**A**, **D**, **E**) or of two independent experiments carried out in duplicates (**C**, **G**) or triplicates (**B**) *p < 0.05, **p < 0.01.

also had a moderate (30%) inhibitory effect on the expression of osteoprotegerin (OPG), a neutralizing decoy receptor of RANKL (*Eriksen, 2010*) (*Figure 1D*). As a result, there was an overall 10-fold decrease in the ratio of OPG/RANKL transcription in galectin-8-treated osteoblasts (*Figure 1E*). We could therefore conclude that galectin-8 increases the RANKL/OPG ratio in osteoblasts in a cell autonomous manner. We have previously shown that galectin-8 is secreted and localizes to the extracellular surface of cells (*Hadari et al., 2000*). To determine whether galectin-8, secreted from osteoblasts, exerts similar effects, the expression of this lectin in osteoblasts was silenced using siRNA. As a result, a reduction of 87% in the expression levels of galectin-8 was accompanied by a significant reduction of 33% in the expression levels of RANKL (*Figure 1F*). We could therefore conclude that galectin-8 derived from osteoblasts can mediate RANKL expression, along with other stimuli that induce RANKL.

To determine whether bone marrow cells can also secrete galectin-8, they were subjected to analysis by flow cytometry. This analysis revealed that indeed primary murine bone marrow cells express and secrete galectin-8 (*Figure 1G*). This surface-bound galectin-8 could be partially displaced by thiodigalactoside (TDG), which blocks lectin–carbohydrate interactions, but not by sucrose, suggesting that surface binding of secreted galectin-8 is mediated, at least in part, through protein–carbohydrate interactions.

## Effects of galectin-8 on RANKL expression in osteoblasts vs osteocytes

Recent studies have implicated matrix-embedded osteocytes, rather than osteoblasts, in the control of osteoclast formation (*Nakashima et al., 2011*; *Xiong et al., 2011*). To determine which cell type serves as a target for galectin-8, calvariae from newborn mice were separated using sequential digestion (*Nakashima et al., 2011*) into osteoblast-rich fraction, expressing the osteoblastogenic marker KERA (keratocan) (*Nakashima et al., 2011*) (*Figure 2A*), and an osteocyte-enriched fraction, which is almost devoid of KERA-expressing cells (*Paic et al., 2009*), but expresses DMP1, an osteocyte marker (*Bonewald, 2011*). As expected, expression of DMP1 was enriched threefold in the kera[(−)] fraction, although a significant number of DMP1[(+)] cells were also present in the kera[(+)] fraction (*Figure 2B*). Basal RANKL expression was much higher in the Kera[(+)] osteoblasts-enriched fraction than in the osteocyte-enriched fraction (*Figure 2C*). Furthermore, the level of RANKL expression in galectin-8-treated cells was fivefold higher than in basal both in Kera[(+)] and in Kera[(−)] cells (*Figure 2C*). Given that the Kera[(−)] fraction was almost completely devoid of osteoblasts, these results support the conclusion that galectin-8 affects RANKL expression both in cultured osteoblasts and osteocytes, derived from calvaria of newborn mice.

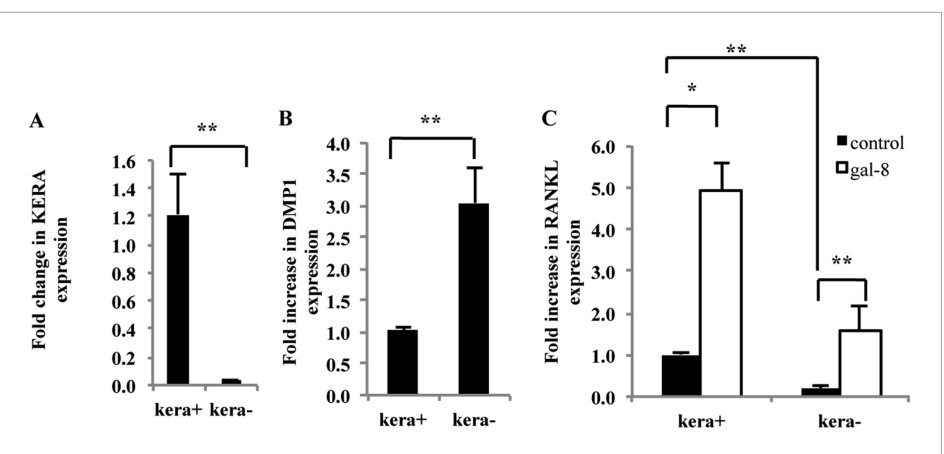

**Figure 2**. Effects of galectin-8 on osteoblast fractions isolated from calvaria of newborn mice. Osteoblasts were extracted from calvaria of newborn mice by five sequential incubations with collagenase-dispase solution. Osteoblasts derived from the different incubations were seeded for 24 hr. KERA (**A**) and DMP1 (**B**) expression, using qRT-PCR, were examined in fractions −2 and −5 that showed the highest and the lowest amount of KERA (designated kera[+] and kera[−]), respectively. (**C**) Galectin-8 (50 nM) was added to osteoblasts from these cultures for 24 hr; RANKL expression was determined by qRT-PCR. Actin served as a control for normalization purposes. Results are mean values ± SEM of n = 8 (**A**), n = 6 (**B**), n = 7 (**C**). *p < 0.05, **p < 0.01.

## Effects of galectin-8 on osteoclasts differentiation

The increased expression of the osteoclastogenic factor RANKL in osteoblasts treated with galectin-8 prompted us to study the effects of this lectin on osteoclasts differentiation in culture. For this purpose, bone marrow cells were co-cultured with osteoblasts derived from calvaria of newborn mice. We could demonstrate (*Figure 3A*) that galectin-8, added to this co-culture, was equipotent to the osteoclastogenic factor $PGE_2$ (*Suda et al., 2004*) in the induction of a ~15-fold increase in osteoclast differentiation, as evident by the appearance of multinucleated $TRAP^+$ cells. The effects of $PGE_2$ and galectin-8 were additive to a certain extent, suggesting that they might act by somewhat different mechanisms. Very few differentiated osteoclasts appeared in untreated co-cultures. Furthermore, galectin-8 had no direct differentiation effect on osteoclasts, as addition of this lectin to naive bone marrow cells in the absence of osteoblasts did not result in osteoclasts differentiation (*Figure 3A*). To verify that RANKL indeed mediates the effects of galectin-8 on osteoclasts differentiation, its expression was silenced using siRNAs. As shown in *Figure 3B*, RANKL-siRNAs reduced its transcription in osteoblasts by 50%, and this was accompanied by a similar 50% reduction in the ability of galectin-8 to induce osteoclastogenesis in co-culture experiments (*Figure 3C*). These results support the conclusion that galectin-8 functions as an osteoclastogenic agent through its action as an inducer of RANKL expression in osteoblasts.

## Signaling pathways triggered by galectin-8 in osteoblasts

The signaling pathways that mediate the effects of galectin-8 on osteoblasts were explored next. We could demonstrate that treatment of osteoblasts, derived from calvaria of newborn mice, with soluble galectin-8 (50 nM, 4 hr), induced the phosphorylation of ERK and Akt, while inhibitors of these signaling pathways—PD98095 and wortmannin, respectively—inhibited these phosphorylations (*Figure 4A*). PD98095 inhibited the ability of galectin-8 to promote transcription of RANKL in osteoblasts, whereas inclusion of wortmannin had no such an effect (*Figure 4B*), suggesting that the effects of galectin-8 on RANKL gene transcription are mediated by the ERK signaling pathway.

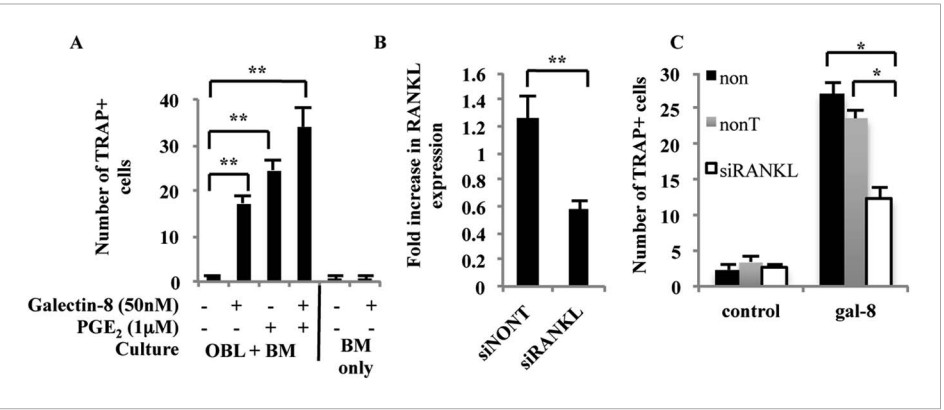

**Figure 3**. Effects of galectin-8 on osteoclast differentiation. (**A**) Osteoblasts (OBL) derived from the calvaria of newborn mice were seeded in 24-well plates ($4 \times 10^4$ cells/well). After reaching 60–70% confluence, murine bone marrow cells (BM) extracted from the femur and tibia of 6-week-old mice were added to the culture ($2 \times 10^6$ cells/well), together with galectin-8 (50 nM), $PGE_2$ (1 µM), or both. Galectin-8 and $PGE_2$ were further added on every other day for 10 days. TRAP assay was performed, and active osteoclasts (multinucleated $TRAP^+$ cells) were counted. Results are mean values ± SEM of three independent experiments carried out in duplicates. (**B**) Osteoblasts were seeded in 12-well plates ($5 \times 10^4$ cells per well). After 24 hr, cells were transfected with siRNA to RANKL. Non-targeting siRNA served as a control. 72 hr thereafter cells were harvested, RNA was extracted, and qRT-PCR was performed in order to quantify changes in mRNA levels of RANKL. The content of actin mRNA served as a control for normalization purposes. (**C**) Osteoblasts were seeded as in **A**. After reaching 60–70% confluence, cells were transfected with the indicated siRNAs for 72 hr. Thereafter, murine bone marrow cells extracted from the femur and tibia bones of 6-week-old mice were added to the culture ($2 \times 10^6$ cells/well). Galectin-8 (50 nM) was added on the first, fourth, and sixth days after addition of bone marrow. Active osteoclasts were counted as in **A**. Results are mean values ± SEM of three (**A**, **B**) and two (**C**) independent experiments each carried out in duplicates (*p < 0.05, **p < 0.01).

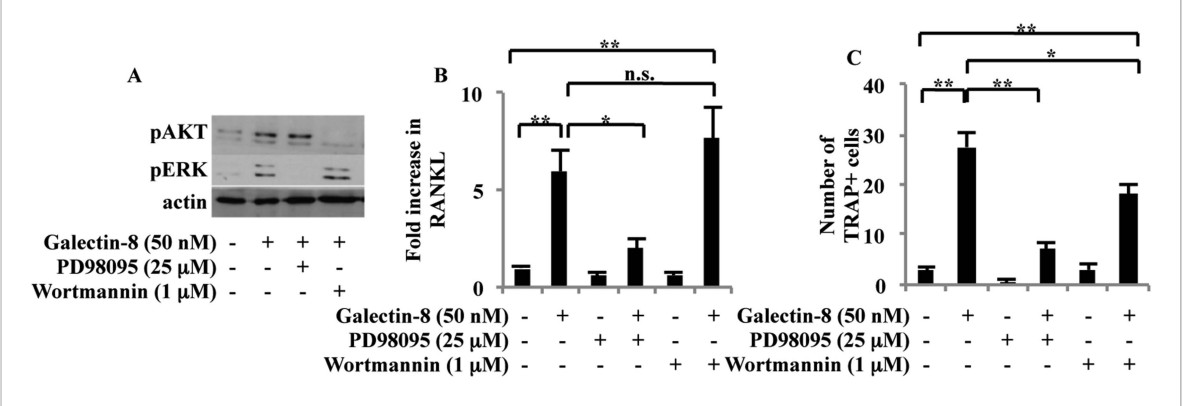

**Figure 4**. Signaling pathways activated by galectin-8. (**A**) Osteoblasts derived from calvaria of newborn mice were treated with 25 μM PD98095 or 1 μM wortmannin for 1 hr before adding galectin-8 (50 nM). After 4 hr, total proteins were extracted and analyzed by Western blotting using antibodies specific for the phosphorylated forms of ERK and Akt. Shown is a representative of three experiments. (**B**) Osteoblasts were treated with PD98095 (25 μM) or wortmannin (1 μM) for 1 hr before being treated with 50 nM galectin-8. After 24 hr, cells were removed from plates, RNA was extracted, and qRT-PCR was performed in order to quantify changes in RANKL transcription. Actin served as a control for normalization purposes. Results shown are mean values ± SEM of three independent experiments, each done in duplicates. (**C**) Osteoblasts were seeded in 24-well plates ($4 \times 10^4$ cells/well). After reaching 60–70% confluence, murine bone marrow cells extracted from the femur and tibia of 6-week-old mice were added to the culture ($2 \times 10^6$ cells/well). Galectin-8 (50 nM), PD98095 (25 μM), and wortmannin (1 μM) were added every other day for 10 days. Multinucleated TRAP+ cells were scored as differentiated osteoclasts. Results shown in (**C**) are mean values ± SEM of two independent experiments each carried out in duplicate. (*p < 0.05, **p < 0.01).

PD98095 also effectively inhibited the appearance of multinucleated TRAP+-differentiated osteoclasts, when bone marrow cells were co-cultured with osteoblasts in the presence of galectin-8 (*Figure 4C*), indicating that the ERK signaling pathway is involved in this process as well. Wortmannin was capable of eliciting a partial inhibitory effect on osteoclast differentiation (*Figure 4C*), suggesting that the PI3K/Akt pathway could act at a step downstream or independent of RANKL transcription.

## Receptors for galectin-8 in osteoblasts

To identify osteoblast receptors that could mediate the effects of galectin-8, proteins extracted from calvaria of newborn rats were affinity purified over columns of immobilized GST-galectin-8 and were analyzed by mass spectrometry. Two proteins that specifically bound to the columns were of interest: LRP1 (low-density lipoprotein receptor-related protein 1) (*Grey et al., 2004*) and MRC2 (mannose receptor C, type 2) (*Engelholm et al., 2009*). Both LRP1 and MRC2 could be detected by staining of proteins that selectively bound to immobilized GST-galectin-8 (*Figure 5A*). Because LRP1 and MRC2 form complexes with the urokinase plasminogen activator receptor (uPAR) (*Behrendt, 2004*; *Gonias et al., 2011*), it was of interest to determine whether uPAR is also part of the proteins complex that binds galectin-8. Indeed, we could show by Western blotting that similar to LRP1 and MRC2, uPAR also selectively binds to immobilized galectin-8 (*Figure 5B*).

To evaluate the possible physiological relevance of these galectin-8-binding partners, their siRNAs were introduced into osteoblasts from calvaria of newborn mice. Transcription of MRC2, LRP1, and uPAR in osteoblasts was reduced >60–80% by their corresponding siRNAs (*Figure 6A–C*). Silencing of MRC2 inhibited (65%) the effects of galectin-8 on RANKL transcription in osteoblasts (*Figure 6D*) and inhibited by 40% the ability of these osteoblasts to promote osteoclastogenesis when co-cultured with bone marrow cells (*Figure 6E*). Similarly, siRNAs to uPAR effectively reduced (50%) the ability of galectin-8 to stimulate expression of RANKL (*Figure 6F*), suggesting that uPAR, like MRC2, mediates at least in part, the stimulatory effects of galectin-8 on RANKL transcription and osteoclastogenesis. By contrast, silencing of LRP1 significantly increased ∼2.5-fold the effects of galectin-8 on RANKL transcription (*Figure 6G*), suggesting that LRP1 could function as an inhibitory decoy receptor for galectin-8, impeding its ability to promote expression of RANKL.

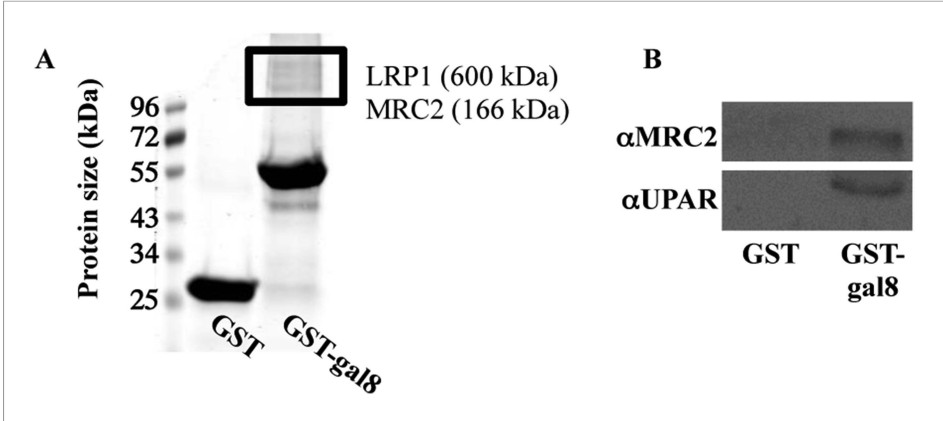

**Figure 5**. Binding of proteins extracted from osteoblasts to GST-galectin-8. (**A**, **B**) Calvariae were isolated from newborn rats (**A**) or mice (**B**); homogenized, and proteins were extracted and incubated for 16 hr at 4°C with GST- or GST-gal-8-loaded beads. Next, the beads were washed in PBS+1% Triton X-100. Elution was performed with 0.5M lactose, and the eluted proteins were resolved by SDS-PAGE and were stained with GelCode (**A**). Relevant bands (marked with a rectangle) were excised, trypsinized, and subjected to analysis by mass spectrometry. Alternatively, the eluted proteins were resolved by SDS-PAGE and were transferred to nitrocellulose membrane for Western blotting with the indicated antibodies (**B**). Blots shown are representatives of four independent experiments with similar results.

## Characteristics of transgenic mice overexpressing galectin-8 (gal-8 Tg)

To further assess the physiological significance of the above findings, transgenic mice that overexpress galectin-8 were generated as described under 'Materials and methods'. These mice express Myc-tagged galectin-8 controlled by the chicken beta-actin promoter that did not include a leader sequence. The insert was localized to chromosome 2, in a region free of genes or other known genomic features. Homozygous mice were used in this study. Mice were born at normal size and expressed no apparent deformity. They were fertile and propagated at a normal Mendelian distribution. Age- and sex-matched mice served as the control group.

Immunohistochemical staining of bone sections and quantitative reverse transcriptase polymerase chain reaction quantitative reverse transcriptase polymerase chain reaction (qRT-PCR) analysis revealed that galectin-8 expression was increased ∼sixfold in osteoblasts derived from calvariae of newborn mice, whereas a ∼10-fold increase was observed in osteoblasts derived from long bones of 16-week-old transgenic mice (Tg) mice, when compared with the control wild-type (WT) mice (*Figure 7A*). Decoration with anti-galectin-8 antibodies of decalcified sections of tibia from 12-week-old WT mice and gal-8 Tg mice confirmed these results. A marked increase in anti-galectin-8 antibody binding was observed in sections of tibia derived from gal-8 Tg mice when compared with WT controls (*Figure 7B*).

## Effects of overexpression of galectin-8 on bone morphology

To determine whether the osteoclastogenic activity of galectin-8 affects bone morphology, indices of tibial bone mass and architecture of WT and Tg mice were determined by micro-computed tomography (μCT) scans both in vivo and in vitro. In vitro μCT of the proximal region of the tibia of 16-week-old mice revealed bone osteopenia in the gal-8 Tg mice (*Figure 8*). This was characterized by a 57% decrease in Tb.N and 62% decrease in BV/TV ratio (*Table 1*). As a consequence, Tb.Sp was significantly higher (2.8-fold) in the Tg animals (*Table 1*). No change in Tb.Th was observed (not shown). A significant reduction (32%) in bone mineral density (BMD) of the Tg mice was observed as well (*Table 1*). Qualitatively, similar changes were also detected by in vivo μCT scans (*Table 1*). Same changes were also evident upon scanning of the distal region of the tibia (not shown).

## Low bone mass in gal-8 Tg mice is accompanied by enhanced bone remodeling

To gain further insight into the mechanisms of reduced bone mass in the gal-8 Tg mice, bone formation parameters were determined by dynamic histomorphometry, using calcein double labeling.

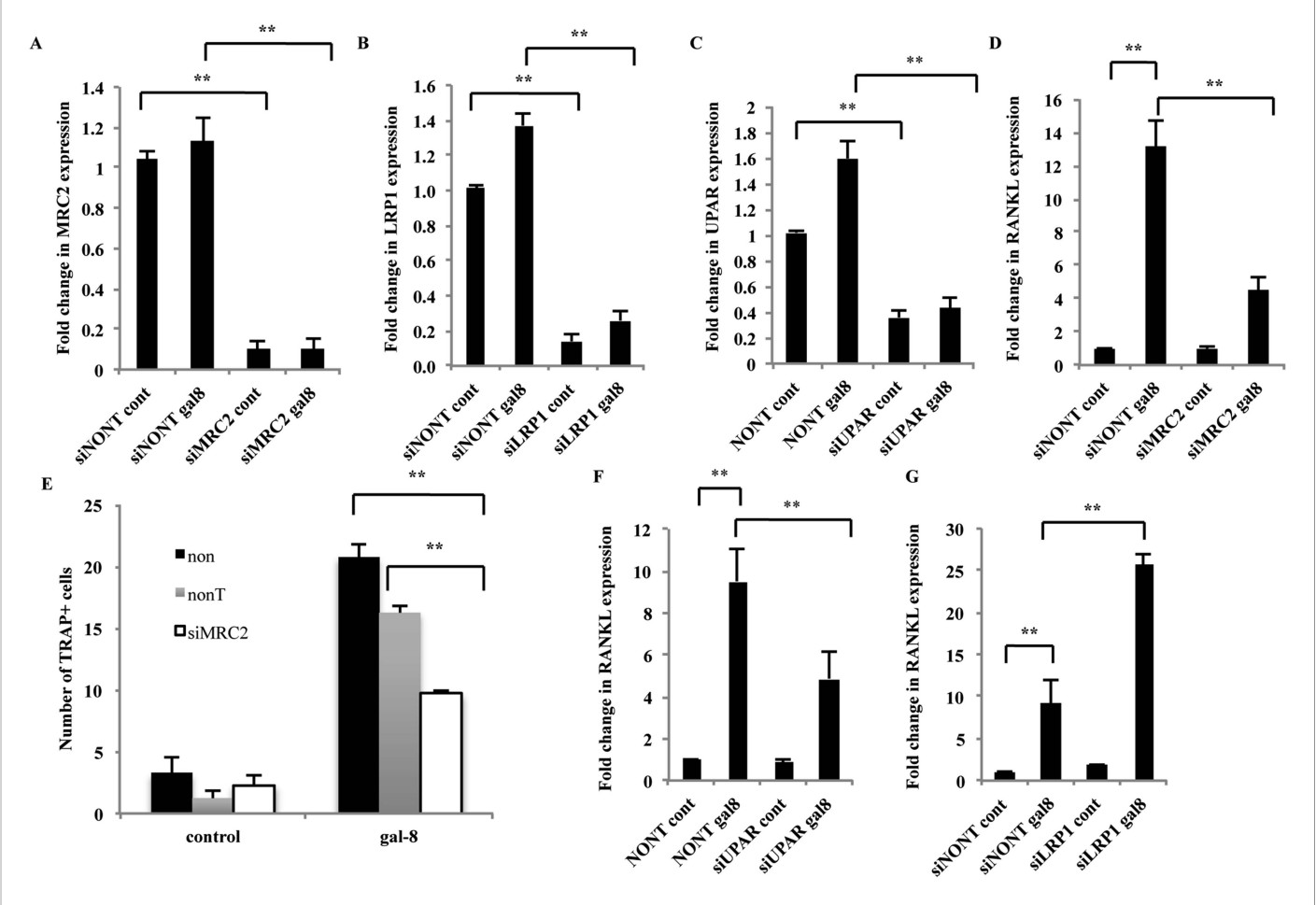

**Figure 6**. Effects of silencing of MRC2, LRP1, and uPAR on the mode of action of galectin-8. (A–D, F, G) Osteoblasts from calvaria of newborn mice were grown in 12-well plates (5 × 10⁴ cells per well). After 24 hr, cells were transfected with the indicated siRNAs. Non-targeting siRNA served as control. 48–72 hr thereafter, galectin-8 (50 nM) was added for another 24 hr. Cells were then harvested, RNA was extracted, and qRT-PCR was conducted to quantify changes in mRNA levels of MRC2 (A), LRP1 (B), uPAR (C), and RANKL (D, F, G). The content of actin mRNA served as a control for normalization purposes. Results shown are mean values ± SEM of (n = 5 [A–C, F, G]; n = 3 [D]) **p < 0.01 vs control. (E) Osteoblasts were seeded in 24-well plates (4 × 10⁴ cells/ well). After reaching 60–70% confluence, cells were transfected with the indicated siRNAs. After 72 hr, murine bone marrow cells extracted from the femur and tibia of 6-week-old mice were added to the culture (2 × 10⁶ cells/well). Galectin-8 (50 nM) was added on the first, fourth, and sixth days after addition of the bone marrow. TRAP assay was performed, and multinucleated TRAP⁺ cells were counted. Results are mean values ± SEM of triplicate measurements repeated in two independent experiments **p < 0.01 vs control.

Mineral appositional rate (MAR), a representation of the activity of the average osteoblast, increased by 25% in the Tg animals (1.12 ± 0.05 µm/day vs 1.41 ± 0.07 µm/day, p < 0.005, *Figure 9*, left). Conversely, bone formation rate (BFR) also increased almost twofold in the gal-8 Tg mice compared with that of the control group (0.30 ± 0.03 µm³/mm²/day vs 0.55 ± 0.06 µm³/mm²/day, p < 0.005, *Figure 9*, right). These results indicate that gal-8 Tg mice experience enhanced bone remodeling that involves enhanced rate of bone formation in spite of overall bone loss in these animals.

## Effects of galectin-8 on RANKL expression and osteoclastogenesis in vivo

To determine whether the increase in BFR in gal-8 Tg mice could be offset by increased osteoclastogenesis in vivo, RNA was extracted from the femur and tibia of 14- to 16-week-old gal-8 Tg mice and WT control animals. As shown in *Figure 10A*, bones of Tg mice expressed 3.3-fold higher amounts of RANKL than those of control WT mice. This was accompanied by a fourfold increase in the

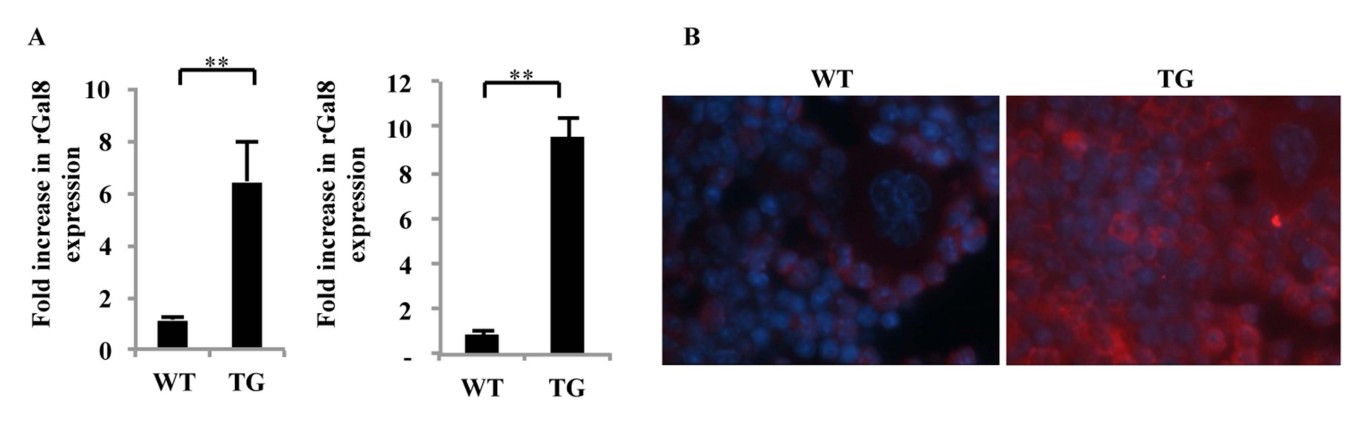

**Figure 7**. Expression of galectin-8 in femur and tibia of gal-8 Tg mice. (**A**) RNA was extracted from osteoblasts derived either from calvaria of newborn mice (n = 7) (left) or from the femur and tibia of 14-week-old (n = 5) (right) wild-type (WT) and gal-8 Tg mice. qRT-PCR was conducted using primers for galectin-8 or actin (control). Result shown are mean ± SEM (**p < 0.01). (**B**) Tibia was removed from 12-week-old WT (left) and gal-8 Tg mice (right). Bones were decalcified and fixed in paraffin blocks. Sections were cut and stained with anti-galectin-8 antibody (red) and DAPI (blue).

ratio of osteoclasts surface/bone perimeter in decalcified sections of tibia from the Tg animals (*Figure 10B,C*) and a 40% increase in the abundance of active osteoclasts in culture (TRAP+, multinucleated cells) (*Figure 10D*). Elevated expression of the osteoclast markers TRAP and cathepsin K (3.5-fold and twofold, respectively) was also observed in bone marrow cells derived from the Tg mice, when compared with their WT controls (*Figure 10E*). No effects of galectin-8 on the transcription of MCSF, the second key osteoclastogenic factor (*Biskobing et al., 1995*), were observed (not shown). Taken together, these results suggest that the increased rate of bone formation in gal-8 -8 Tg mice is apparently offset by the even greater increase in osteoclastogenic activity in the bones of the Tg animals, resulting in a net loss of bone mass.

## Discussion

Excessive reduction in bone mass, with osteoporosis as one of its hallmarks, is a major health problem, especially in the elderly population (*Manolagas and Jilka, 1995*). Still, the mechanisms underlying the development of this disease and some of the critical players in this process remain partially obscure (*Bonucci and Ballanti, 2014*). In the present work, we provide evidence that an animal lectin of the galectin family regulates osteoclastogenesis and loss of bone mass. We show that galectin-8 increases the expression of RANKL, a key osteoclastogenic factor (*Hanada et al., 2011*; *Xiong and O'Brien, 2012*), in cultured osteoblasts, and promotes their osteoclastogenic potential when co-cultured with bone marrow cells. At the same time galectin-8 inhibits the expression of OPG, a neutralizing decoy receptor of RANKL, thus leading to an overall increase in the RANKL/OPG ratio. Given that RANKL is both necessary and sufficient for osteoclast differentiation, provided that permissive concentrations of MCSF are present (*Eghbali-Fatourechi et al., 2003*), and given that galectin-8 does not affect the expression of MCSF, our findings strongly suggest that RANKL mediates the potentiating effects of galectin-8 on bone resorption.

Galectins are secreted by atypical secretory pathway from different cell types (*Boscher et al., 2011*; *Di Lella et al., 2011*). Therefore, a number of cell types could serve as a source for the secreted galectin-8 that induces RANKL expression by osteoblasts. We have shown that silencing of galectin-8 expression in osteoblasts partially inhibits the expression of RANKL, suggesting that galectin-8, secreted from these cells, could act in an autocrine fashion to induce RANKL expression. However, we could also show that galectin-8 is present on the surface of bone marrow cells, turning these cells into additional potential source for galectin-8 that affects osteoblasts. Further studies will be required to resolve this complex issue.

At the molecular level, galectin-8 binds at the cell surface of osteoblasts to the low-density lipoprotein receptor-related protein 1 (LRP1) (*Grey et al., 2004*) and the mannose receptor C, type 2

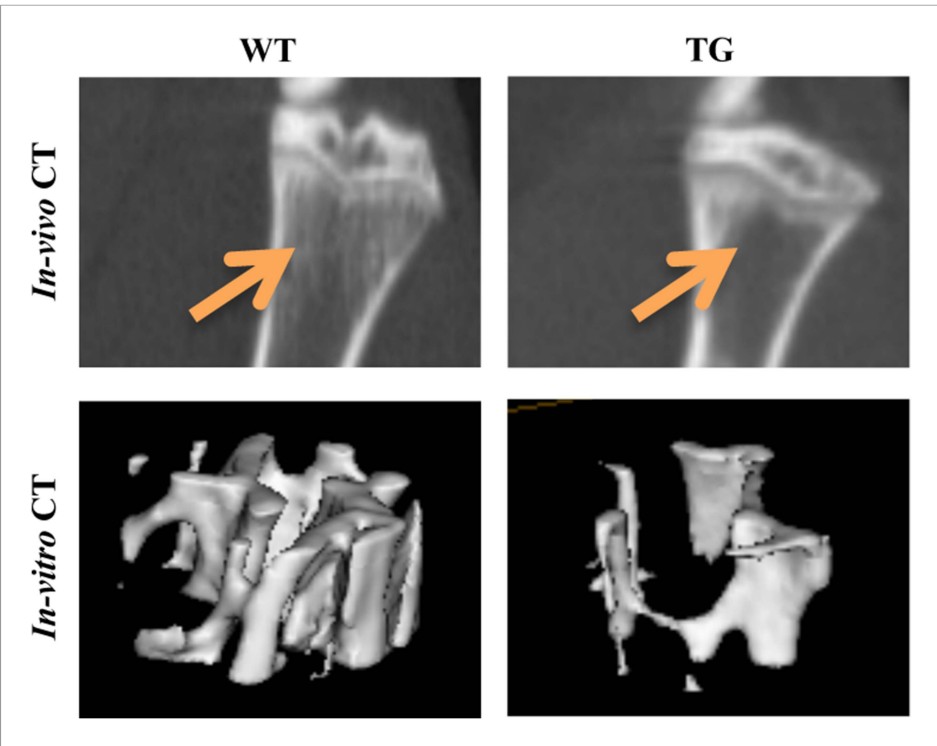

**Figure 8**. MicroCT scans of tibia proximal regions. In vivo and in vitro μCT scans were performed on 14-week-old WT and Tg mice, or on tibial bones removed from 16-week-old WT and Tg mice, respectively. Representative pictures show the proximal region of the tibia (for in vivo CT) or a region of interest within the trabecular bone of the tibia metaphysis (for in vitro CT). The position of Trabeculae is indicated by arrows.

(MRC2) (*Engelholm et al., 2009*) that negatively and positively regulate galectin-8 function, respectively. LRP1 and MRC2 are part of a multi-protein complex that includes uPAR (*Behrendt et al., 2000*; *Smith and Marshall, 2010*). Indeed, we could demonstrate that uPAR co-precipitates with galectin-8 and that partial silencing the expression of uPAR attenuates the ability of galectin-8 to promote RANKL transcription; findings that conform with the observation that uPAR-deficient mice have increased bone mass (*Furlan et al., 2007*). uPAR localizes to integrin-containing adhesion complexes, and co-immunoprecipitates with integrins and integrin-associated signaling molecules such as focal adhesion kinase (FAK) and Src family kinases (reviewed in *Smith and Marshall, 2010*). In particular, it modulates the affinity of $\beta_1$, $\beta_2$, and $\beta_3$ integrins (*Wei et al., 1996*) for their

**Table 1**. Analysis and stereological parameters of tibia proximal region in WT and gal-8 Tg mice

| | In vivo CT | | In vitro CT | |
|---|---|---|---|---|
| | WT (n = 7) | Tg (n = 7) | WT (n = 5) | Tg (n = 5) |
| BV/TV | 0.40 ± 0.02 | 0.25 ± 0.03** | 0.12 ± 0.01 | 0.04 ± 0.01** |
| Tb.N (1/mm) | 3.19 ± 0.02 | 1.86 ± 0.25** | 3.86 ± 0.51 | 1.68 ± 0.34** |
| Tb.Sp (mm) | 0.19 ± 0.01 | 0.48 ± 0.08** | 0.24 ± 0.03 | 0.68 ± 0.18* |
| BMD (%) | 100% ± 15% | 52% ± 5%** | 100% ± 8% | 68% ± 6%** |

14-week-old WT and Tg mice (n = 7 each group) were scanned using a small animal in vivo μCT scanner. Tibial bones were removed from 16-week-old WT and Tg mice (n = 5 each group) and scanned using an in vitro CT scanner. Analysis was performed on the proximal region of the tibia. The parameters calculated are Tb.N (trabecular number), Tb.Sp (trabecular separation), BV/TV (bone volume/tissue volume), and BMD (bone mineral density). Results shown are mean values ± SEM. BMD is given as relative to the average BMD of WT mice (**p < 0.01 vs WT mice).

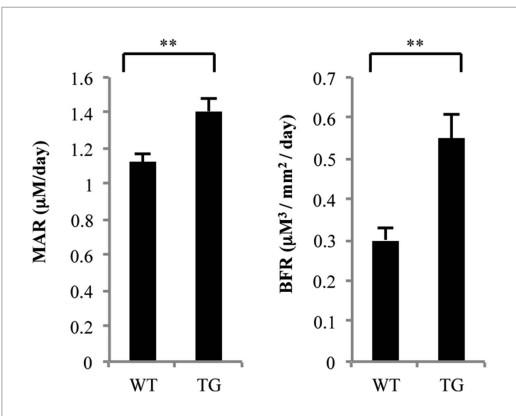

**Figure 9**. Histomorphometric analysis of femurs from WT and gal-8 Tg mice. 16-week-old WT (n = 5) and Tg (n = 9) mice were injected with calcein. MAR (mineral apposition rate, left) and BFR (bone formation rate, right) were measured and calculated based on sections of the femur bone taken from these mice. Results shown are mean values ± SEM (**p < 0.01 vs WT mice).

corresponding matrix ligands (*Smith and Marshall, 2010*), and does it, in part, through binding to vitronectin (*Smith and Marshall, 2010*). Integrins, including $\alpha_1$, $\alpha M$, $\alpha_3\beta_1$, and $\alpha_6\beta_1$, as well as ECM proteins, also serve as binding partners to galectin-8 that functions as a matricellular protein (*Hadari et al., 2000*; *Nishi et al., 2003*; *Cárcamo et al., 2006*; *Troncoso et al., 2014*). Complex formation between galectin-8 and integrins triggers integrin-mediated signaling cascades such as Tyr phosphorylation of FAK and paxillin, and a robust and sustained activation of the ERK and PI3K pathways (*Levy et al., 2001*, *2003*). Hence, interaction of galectin-8 with a complex of the uPAR/LRP1/MRC2 that binds integrins could be the mechanism underlying RANKL transcription in osteoblasts treated with this lectin. This model is further supported by the fact that integrins promote RANKL transcription through the activation of FAK in osteoblasts (*Nakayamada et al., 2003*). Phosphorylated FAK activates several transduction molecules including Src and Grb2, which activate the ERK and PI3K signaling pathways (*Schwartz and Ginsberg, 2002*). Indeed, activation of the ERK pathway in osteoblasts is obligatory for the action of galectin-8 as an inducer of osteoclastogenesis, implicating this signaling pathway as being the major pathway activated downstream of galectin-8. In contrast, the PI3K pathway appears not to play a role in the induction of RANKL in response to galectin-8, while having a partial role in mediating osteoclastogenesis. These findings place the PI3K pathway as acting downstream or independent of RANKL transcription in mediating the effects of galectin-8 on osteoclastogenesis.

The mechanism underlying the inhibitory effects of LRP1 on RANKL transcription in response to galectin-8 is presently unknown. LRP1, which also directly interacts with uPAR (*Gonias et al., 2011*), could, for example, facilitate the endocytosis of uPAR, which was the first cell-signaling receptor identified as a member of the LRP1-regulated plasma membrane proteome (*Gaultier et al., 2006*). Because LRP1 down-regulates cell-surface uPAR by facilitating its endocytosis, uPAR-initiated cell signaling may be inhibited by LRP1. Although galectin-8 can potentially bind to receptors that both negatively (LRP1) and positively (MRC2 and uPAR) regulate RANKL expression, the net effect is still an increase in RANKL expression and reduction in bone mass, which could be attributed, for example, to different expression levels of these receptors in osteoblasts. Alternatively, the duration of the signals emitted by these receptors could differ.

Osteoblastic cells are considered as the major cell type that expresses RANKL to support osteoclastogenesis; however, recent findings suggest that osteocytes are the main regulators of bone homeostasis through RANKL expression (*Nakashima et al., 2011*; *Xiong et al., 2011*). Our findings reveal that galectin-8 affects RANKL expression in osteoblasts-enriched fractions of calvariae as well as in osteocyte-enriched fraction. These findings support the hypothesis that galectin-8 acts both on osteoblasts and osteocytes, at least in culture, and still, we cannot rule out the possibility that it induces RANKL expression in vivo selectively in osteocytes.

The increased expression of RANKL in gal-8 transgenic animals enhances their osteoclasts differentiation and results in a reduction in their BMD and bone volume fraction (BVF), thus supporting the notion galectin-8 could induce a physiological bone loss. Of interest, dynamic histomorphometry revealed active bone remodeling, associated with increased rate of bone formation in gal-8 Tg mice. The increase in bone turnover could be attributed to the enhanced osteoclastogenic activity induced by galectin-8 that subsequently promotes osteoblasts differentiation and increased BFR (*Feng and McDonald, 2011*; *Bonucci and Ballanti, 2014*). Still, the net effect is bone loss due to the overall greater activity of the osteoclasts. In this respect, our model of Tg mice resembles in certain aspects the changes in bone turnover that take place during postmenopausal osteoporosis (*Raisz, 2005*; *Feng and McDonald, 2011*). It is well established

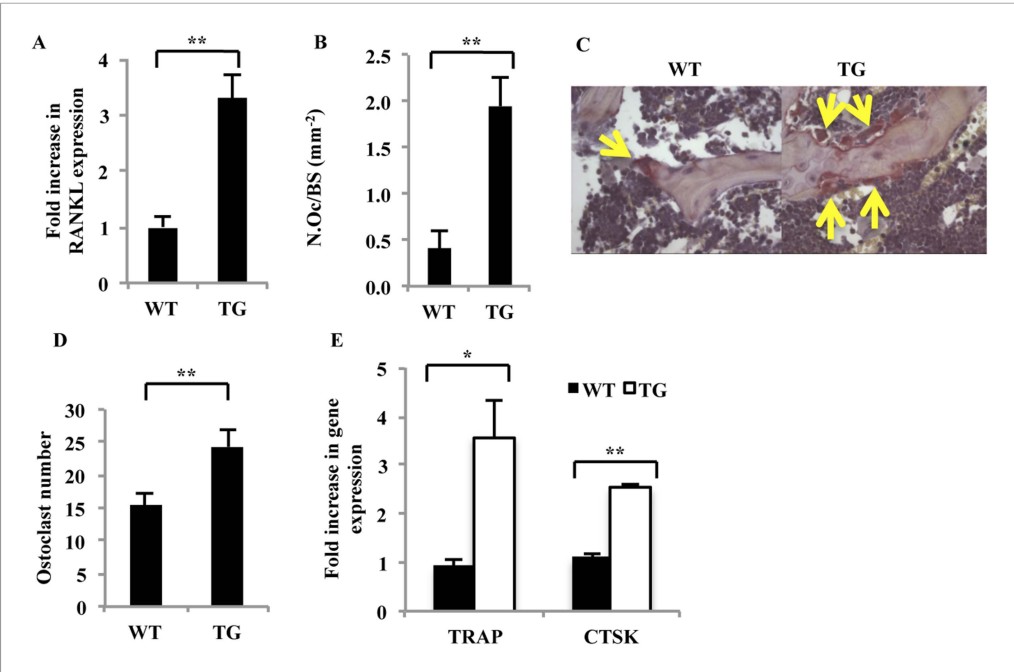

**Figure 10**. Characterization of osteoblasts and osteoclasts derived from WT and gal-8 Tg mice. (**A**) RNA was extracted from the femur and tibia of 14–16-week-old WT and Tg mice. qRT-PCR was conducted in order to quantify changes in expression of RANKL. Actin served as a control for normalization purposes. Results are mean values ± SEM of five mice per group. (**B**) Femurs were removed from 16-week-old WT and gal-8 Tg mice. Bones were fixed, and TRAP staining was performed. Quantification of the ratios of osteoclast number to bone surface was calculated from sections of WT (n = 5) and Tg (n = 9) mice. Results are mean values ± SEM (**p < 0.01). Representative sections are shown in (**C**) Arrows indicate the position of osteoclasts. (**D**, **E**) Bone marrow cells were extracted from the femur and tibia bone of 14-week-old WT and Tg mice and were seeded in 24-well plates ($2 \times 10^6$ cells/well) for 24 hr. The number of multinucleated TRAP+ cells was determined (**D**), and qRT-PCR of the indicated genes was performed (**E**). Results are mean values ± SEM of n = 6 and n = 4 mice/group in (**D**) and (**E**), respectively (*p < 0.05, **p < 0.01 vs WT cells).

that both bone resorption and BFRs are increased during postmenopausal osteoporosis (*Raisz, 2005*; *Feng and McDonald, 2011*); however, the extent of increased bone resorption exceeds that of augmented bone formation, which causes an imbalance in favor of bone loss (*Arlot et al., 1990*; *Ebeling et al., 1996*; *Tanizawa et al., 1999*; *Raisz, 2005*; *Eriksen, 2010*).

A number of rodent models for postmenopausal osteoporosis exist (e.g., *Erlebacher and Derynck, 1996*; *Bucay et al., 1998*; *Mizuno et al., 2002*; *Rinotas et al., 2014*). One is a model of Tg mice that overexpress human RANKL (*Rinotas et al., 2014*). These mice develop bone loss and increased bone turnover rate, which is similar to the phenotype of gal-8 Tg mice, supporting the notion that galectin-8 mainly acts through increased expression of RANKL in vivo.

In summary, our findings implicate an animal lectin as a novel regulator of osteoclastogenesis and bone remodeling. The unique aspect of these observations stems from the fact that galectin-8, like other secreted animal lectins, binds cell-surface glycoconjugates that enable it to engage in binding to a number of receptors that express the proper repertoire of sugars on their surface. This offers a novel mean for spatially controlled regulation of bone remodeling through high-density information coding that involves lectin–sugar interactions. Our findings place animal lectins, with galectin-8 as their representative, as novel osteoclastogenic agents and regulators of bone remodeling. The phenotype of gal-8 Tg mice, showing enhanced bone turnover and reduced bone mass, supports the conclusion that galectin-8 induces RANKL expression in an in vivo setting. The results reveal a potential link between galectins, LRP1, MRC2, and uPAR in mediating this process of osteoclastogenesis. Therefore, insights from this study might inform efforts to develop novel drug targets for the treatment of diseases associated with bone loss.

## Materials and methods

### Materials

Commercially available reagents were purchased from the following resources: trypsin-Ethylendiaminetetraacetic acid (EDTA), penicillin, and L-glutamine were purchased from Biological Industries (Beit Haemek, Israel). Fetal bovine serum was obtained from Hyclone Laboratories Inc. (Logan, UT). Isopropyl-β-D-thiogalactopyranoside (IPTG) and PCR Master Mix (Dreamtaq) were purchased from MBI Fermentas (Amherst, NY). Dispase II (neutral protease, grade II) was from Roche Diagnostics (Mannheim, Germany). siRNA SMARTpool libraries were provided by Dharmacon (Lafayette, CO, USA). GST-coupled resins were purchased from Novagen (Madison, WI). Lipofectamine 2000 was from GIBCO-BRL (Grand Island, NY). PerfectPure RNA Cell & Tissue for RNA extraction was from 5 PRIME (Hamburg, Germany). cDNA reverse transcription kit was purchased from Applied Biosystems (Carlsbad, CA). Real-time Polymerase Chain Reaction (PCR) kit (SYBR green PCR master mix) was purchased from Invitrogen (Carlsbad, CA). Thiodigalactoside was purchased from Santa Cruz Biotechnology (Dallas, TX). Leukocyte acid phosphatase staining kit, protease inhibitor cocktail, lactosyl-sepharose beads, wortmannin, cycloheximide, proteinase K, lysozyme, collagenase type 1A, prostaglandin $E_2$, Dulbecco's Modified Eagle Medium (DMEM), sucrose, and diethyl pyrocarbonate (DEPC) were purchased from Sigma Chemicals Co. (St. Louis, MO). Galectin-8 was a bacterially expressed recombinant protein, encoded by the cDNA of rat galectin-8 (*Hadari et al., 1995*). Monoclonal galectin-8 antibodies (106.1) were generated as described (*Levy et al., 2001*).

### Mice

CB6F1 mice were used throughout this study. All animals were housed under standard light/dark conditions in the animal care unit of the Weizmann Institute of Science. Mice were given food and water ad libitum. Experiments were approved by the Animal Care and Use Committee of the Weizmann Institute of Science.

### Generation of transgenic mice

Plasmid-bearing Myc-tagged rat galectin-8 coding sequence under the chicken beta-actin promoter was constructed on the backbone plasmid pQE-TrySystem. The plasmid was restricted by NaeI and SphI, and the linear fragment of 3137 bp was microinjected into CB6F1-fertilized oocytes. Mice were scanned for the presence of the insert by two PCR reactions using two pairs of primers for amplification of the promoter region (sense: AAAGGAGATATACCGCGGCGA TATCCC, antisense: CTGCAACCTTGAACTCTCGGACATCAC, 630 bp) and the myc-galectin-8 coding sequence (sense: CGCCAATAGGGACTTTCCATTGAC, antisense: CTAATTACAGCCC GAAGGAGAAGG, 970 bp).

### Osteoblast isolation from murine calvariae

Isolation and culture of osteoblasts from newborn mice calvariae were carried out as described (*Bakker and Klein-Nulend, 2003*; *Nakashima et al., 2011*). Briefly, calvariae were extracted from up to 1-day-old pups and were subjected to five incubations with 0.1% collagenase and 0.2% dispase in serum-free medium. The medium from the second to the fifth incubations were collected and centrifuged to pellet the osteoblasts. Osteoblasts were then grown until sub-confluence and were frozen till further use. For each experiment, cells were thawed and grown in α-modified DMEM medium (Sigma) supplemented with 10% fetal calf serum.

### Osteoclast differentiation assay

The ability of osteoblasts to induce osteoclast differentiation was assayed in a co-culture system, adapted from *Takahashi et al. (2007)*. Primary osteoblasts ($1 \times 10^4$ cells/well) extracted from calvariae of newborn mice were co-cultured in 24-well plates with bone marrow cells ($1 \times 10^6$ cells/well) flushed from the femur and tibia of 6-week-old mice. Medium, containing galectin-8 and/or $PGE_2$, was replaced every other day. After 10 days, cells were stained for TRAP (tartarate-resistant acid phosphatase) using a commercial staining kit (Sigma) following the manufacturer instructions. Multinucleated cells stained purple were counted as active osteoclasts.

## RNA analysis

Cells were grown in 6-well plates. Following treatment, cells were harvested and total RNA was extracted using the PerfectPure RNA kit (5 PRIME). RNA was quantified and cDNA was generated by cDNA Reverse Transcription kit (Applied Biosystems) following manufacturer instructions. Quantitative detection of mRNA transcripts was carried out by real-time PCR using ABI-Prism 7300 instrument (Applied Biosystems) using SYBR Green PCR mix (Invitrogen) and specific primers (400 nM final concentration) (*Table 2*). Results were normalized to mRNA levels of β-actin.

## RNA extraction from murine long bones

Femur and tibia bones were removed and placed in RNAlater solution (Ambion, Foster City, CA). After flushing of the bone marrow, and two more flushings with RNAlater solution, bones were cut into 1- to 2-mm$^3$ pieces and crushed. Total RNA was extracted as detailed above.

## Western blot analysis

Cells were harvested in lysis buffer (25 mM Tris/HCl, 25 mM NaCl, 0.5 mM Ethylene Glycol Tetraacetic Acid (EGTA), 2 mM sodium orthovanadate, 10 mM NaF, 10 mM sodium pyrophosphate, 80 mM β-glycerophosphate, 1% Triton X-100, 0.05% Sodium Dodecyl Sulphate (SDS) and protease inhibitors 1:1000, pH 7.5) and were centrifuged at 12,000×$g$ for 20 min at 4°C. Supernatants were collected, and samples of 50 µg protein were mixed with 5×Laemmli sample buffer and were resolved by SDS-PAGE under reducing conditions. Proteins were transferred to nitrocellulose membrane for Western blotting with the indicated antibodies.

## Quantification of soluble RANKL

Media from cells were used for quantification of soluble RANKL using a murine sRANKL ELISA Development Kit (PeproTech) according to the manufacturer instructions. sRANKL levels were normalized to total cellular protein concentration, quantified by Bradford assay.

## Flow cytometry analysis

Murine bone marrow cells were extracted and washed with 10 mM TDG or sucrose for 15 min at 4°C. Flow cytometry was performed as previously described (*Isaac et al., 2013*). Briefly, after 20 min of fixation with 2% *p*-formaldehyde (PFA), cells were incubated on ice with 0.5% BSA for 30 min, followed by incubation with galectin-8 antibodies for 1 hr and Alexa 594-labeled secondary antibodies (Life Technologies) for 30 min. Cells were washed with cold phosphate-buffered saline (PBS) between incubations. Flow cytometry analysis was performed by LSR II Flow Cytometer System (BD Biosciences).

## Histological staining of bones

The tibia and femur were removed from 14- to 16-week-old mice and cut open at their distal end. Fixation was performed in 2.5% PFA for 48 hr, followed by decalcification in 10% EDTA for 3 days. Sections were stained with anti-galectin-8 antibodies or with TRAP staining kit (Sigma) following manufacturer instructions. Measurements were performed on blindly selected regions taken from each slide.

**Table 2**. qRT-PCR primer sequences (5′–3′)

| Gene | Forward primer | Reverse primer |
| --- | --- | --- |
| RANKL | ATCGGGAAGCGTACCTACAG | GTGCTCCCTCCTTTCATCAG |
| TRAP | CAGCAGCCAAGGAGGACTAC | ACATAGCCCACACCGTTCTC |
| CTSK | CAGCTTCCCCAAGATGTGAT | AGCACCAACGAGAGGAGAAA |
| MRC2 | GCCATACGGCTTTGCCCTAC | GGCCCTGGATTCGGAAACAC |
| uPAR | TGTGCTGGGAAACCGGAGTT | GAGGTGGGTCGGGAAGGAGT |
| LRP1 | TCAGACGAGCCTCCAGACTCT | ACAGATGAAGGCAGGGTTGGT |
| Gal-8 (mouse) | TGAACACCAATGCCCGAAGC | GCGTGGGTTCAAGTGCAGAG |
| Gal-8 (rat) | TGTATGCCCACAGGATCAAC | ATCCGAGCTGAATCTGAACC |

## Bone mass and structure analysis by µCT

For in vitro CT analysis, the tibia was removed from 16-week-old mice and scanned using an in vitro µCT scanner (MicroXCT-400, Xradia, California, USA). For each bone, 300 projection images were taken over 180°, with an exposure time of 3 s per projection and geometry set for a voxel size of 5.67 µm (source-to-sample distance 90 mm, sample-to-detector distance 17 mm, linear magnification 4×). All morphometric parameters were determined by using a direct 3D approach. Volume reconstruction was performed with a dedicated software (Xradia California, USA) based on the filtered back-projection algorithm. Parameters determined in the metaphyseal trabecular bone included bone volume density (BV/TV), trabecular number (Tb.N), and trabecular thickness (Tb.Th) and were estimated using a region of interest (ROI) size of $150 \times 150 \times 50$ voxels, which was placed in the center of the trabecular region of the tibia, approximately 300 µm below the lowest point of the growth plate.

For in vivo CT analysis, mice were anesthetized by injection of 2% ketamine and xylasine in PBS. Mice were scanned using small animal in vivo µCT scanner (TomoScope 30S duo, VAMP, Germany) following instrument-operating instructions. Scans were performed using the 65-65-360-90 protocol (using two micro-focus x-ray tubes of 65 Kv with an integration time of 90 ms), with a resolution of 80 µm. Image reconstruction was carried out by the Impact View software (VAMP, Germany). Files were saved in digital imaging and communications in medicine (DICOM) format. Conversion of DICOM files to analysis files was carried out using imageJ software (National Institutes of Health). BVF, BMD, and stereological bone parameters were calculated using the eXplore MicroView software (GE Healthcare, UK). Calculations were performed in a cylinder-shaped ROI, with a size of $10 \times 10 \times 10$ voxels, which was placed inside the trabecular region of the proximal tibia, in similar positions in all mice. For BVF analysis, the automated threshold function of MicroView was used for bone segmentation.

## Histomorphometric analysis

To label bone-forming surfaces, mice were injected subcutaneously with calcein (Sigma Chemical Co, St. Louis, MO) at 15 mg/kg, 4 and 1 day before sacrifice. After sacrifice, femurs were removed and kept in 70% ethanol until µCT and histomorphometric analyses. After µCT image acquisition, femora were embedded undecalcified in polymethylmethacrylate (Technovit 9100; Heraeus Kulzer, Wehrheim, Germany) (*Parfitt et al., 1987*). Longitudinal 5-µm sections were employed for dynamic histomorphometric measurements based on calcein labeling, using a fluorescent microscope. The analysis was carried out in a blinded manner on digital photomicrographs using IMAGE-PRO EXPRESS 4.0 image analysis software (Media Cybernetics, Silver Spring, MD). The following parameters were determined in a reference area extending 0.75–2.25 mm proximal of the distal growth plate in a mid-longitudinal plane according to the convention of standardized nomenclature (*Parfitt et al., 1987*): total bone surface (BS), the percentage of single- and double-labeled bone surfaces (sLS and dLS), and interlabel width were measured. Mineralizing surface [MS/BS = (dLS + 1/2sLS)/BS], mineral apposition rate [MAR = interlabel width/labeling time interval], and BFR [= MAR (MS/BS)] were calculated according to convention. To determine osteoclast number, consecutive sections were deplasticized, and TRAP (tartrate-resistant acid phosphatase) staining was used (Sigma, St. Louis, MO, USA). Stained osteoclasts were counted, and their numbers were determined per millimeter of trabecular bone surface (Oc.N/BS) in the same reference area as previously described by us (*Dresner-Pollak et al., 2008*).

## siRNA transfections

Osteoblasts were transfected with siRNA SMARTpool (Dharmacon) by using Lipofectamine 2000 transfection reagent (GIBCO-BRL, Grand Island, NY) according to manufacturer's instructions. Briefly, 50 pmol of the siRNA was diluted in 50 µl of serum-free medium. 1 µl of Lipofectamine 2000 was also diluted in the same volume of serum-free medium. Both solutions were mixed together and incubated at room temperature for 15 min. The mix was added at 1:10 ratio to the osteoblasts' culture medium. Incubation with the siRNA was carried out for 48 hr.

## Identification of galectin-8-binding partners in osteoblasts by mass spectrometry

Calvariae were extracted from newborn rats and homogenized, and proteins were extracted as described above. Glutathione S-transferase (GST) and GST-galectin-8 (100 µg) were immobilized on 200 µl glutathione–agarose beads at 4°C. After 2 hr, the beads were washed three times with PBS. Calvaria proteins were incubated for 16 hr with GST- or GST-gal-8-loaded beads. The beads were

loaded on a column and washed with 100 ml of PBS in Triton X-100 (0.1%) and protease inhibitors. Proteins were eluted with 0.5M lactose supplemented with 0.1% Triton and protease inhibitors cocktail. Eluted proteins were resolved by SDS-PAGE and were stained with Gel Code. Protein bands that were selectively eluted from GST-gal-8-loaded beads in two independent experiments were excised and trypsinized. Resulting peptides were subjected to mass spectrometry analysis and nano-LC-ESI-MS/MS as we previously described (*Isaac et al., 2013*). For the analysis of tryptic peptides, survey scans were recorded in the Fault-tolerance (FT) -mode followed by data-dependent collision-induced dissociation of the seven most-intense ions in the linear ion trap. The data were searched with MASCOT (Matrix Science, London, UK) against a Swissprot or National Center for Biotechnology Information (NCBI) databases and confirmed by manual inspection of the fragmentation series. Relative quantitation was conducted with the Scaffold PTM software (Proteome Software Inc., USA) and the A-score algorithm (*Zhai et al., 2008*).

## Statistics
Data are presented as mean ± SEM unless otherwise specified. Group means were compared using the non-paired t-test. Differences of $p < 0.05$ were considered significant.

## Acknowledgements

We thank S Sampson (Bar Ilan University, Israel) and E Elhanany (Israel Institute for Biological Research) for a critical review of the manuscript and Shifra Ben-Dor for help in bioinformatics analysis. This work was supported by a research grants from the WIS Center for Women Health, The Joseph Cohen Minerva Foundation (711425), and the Israel Science Foundation (776/13). Y Zick is the incumbent of the Marte R Gomez professorial chair. This work is dedicated to the memory of Itai Bab.

## Additional information

### Funding

| Funder | Grant reference | Author |
| --- | --- | --- |
| Weizmann Institute of Science | Center for Women Health | Yehiel Zick |
| Minerva Foundation (Minerva Stiftung) | 711425 | Yehiel Zick |
| Israel Science Foundation (ISF) | 776/13 | Yehiel Zick |

The funders had no role in study design, data collection and interpretation, or the decision to submit the work for publication.

### Author contributions

YV, VB, SB-H, Conception and design, Acquisition of data, Analysis and interpretation of data, Drafting or revising the article; HS-A, Conception and design, Acquisition of data, Analysis and interpretation of data; AV, NH, YL, RK, SB, MA-L, Acquisition of data, Analysis and interpretation of data; IB, YZ, Conception and design, Analysis and interpretation of data, Drafting or revising the article

### Ethics
Animal experimentation: All of the animals were handled according to approved Institutional Animal Care and Use Committee (IACUC) protocols (#00950212-2) of the Weizmann Institute of Science. The protocol was approved by the Committee on the Ethics of Animal Experiments of the Weizmann Institute. The IACUC, appointed by the President of the Weizmann Institute, reviews all experimental protocols in which experimental animals are required. Furthermore the committee reviews and approves the generation and use of genetically modified animals. The Committee makes sure that experimental animals of the species and the numbers requested are needed for the proposed research. Moreover, all procedures employed on experimental animals, including their transportation, routine care and use in experiments are conducted in accordance with the Israel animal welfare law and guidelines, NIH guidelines and the Animal Welfare Act, the ethical standards and guidelines of FP7, with the EU directive 86/609/EEC as well as the revised directive 2010/63/EU on the protection of animals used for scientific purposes.

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
