## [Decision Letter]

Thank you for sending your work entitled “The mammalian lectin, galectin-8, induces RANKL expression, osteoclastogenesis and reduction in bone mass of mice” for consideration at *eLife*. Your article has been favorably evaluated by Janet Rossant (Senior editor), a Reviewing editor, and three reviewers, one of whom, Toshio Kukita, has agreed to share his identity.

The Reviewing editor and the reviewers discussed their comments before we reached this decision, and the Reviewing editor has assembled the following comments to help you prepare a revised submission.

The three reviewers and the editors concurred that your work represents a novel demonstration of the effects of galectin-8 on osteoclastogenesis, with the finding that galectin-8 exhibits both positive (MRC-2 and uPAR) and negative (LRP1) regulation. The reported effects could be significant for further advancing our understanding of bone metabolism.

Major concerns raised and suggestions for revision:

1) The authors should explicitly address the observation that, while galectin-8 binds to receptors that both negatively and positively regulate osteoclastogenesis, the net effect in galectin-8 transgenic mouse is the loss of bone mass.

2) Galectins are present in the cytosol, nucleus and the extracellular space, and they can bind to many cell types. For galectins present in the extracellular space, it is not clear whether they function as suggested by using recombinant protein added to single cell types. It is thus critical to demonstrate the function of endogenous galectins.

3) A key question for understanding the results of this work is whether galectin-8 is expressed in the bone marrow and whether it is secreted and thus present on the cell surfaces. The authors should determine whether galectin-8 could be eluted by treatment with lactose (and not control saccharides).

4) If the answer to question #3 is negative, the question is whether the described effects of galectin-8 can in fact occur naturally. If so, the authors need to demonstrate the function of endogenous galectin-8 by using relevant cells with galectin-8 expression knocked down or using cells from galectin-8 knockout mice. The use of transgenic mice overexpressing galectin-8 might be problematic, as it represents an artificial situation with the amounts of galectin-8 released into the environment that might not be reached in wild-type mice.

---

## [Author Response]

*1) The authors should explicitly address the observation that, while galectin-8 binds to receptors that both negatively and positively regulate osteoclastogenesis, the net effect in galectin-8 transgenic mouse is the loss of bone mass*.

Although galectin-8 can bind to receptors that either negatively (LRP1) or positively (MRC2 and uPAR) regulate RANKL expression, the net effect is still an increase in RANKL expression and reduction in bone mass. This could be attributed, for example, to different expression levels of these receptors in osteoblasts. Alternatively, the duration of the signals emitted by these receptors could differ. We have addressed this point in the Discussion (“Although galectin-8 can potentially bind to receptors […] the signals emitted by these receptors could differ”).

*2) Galectins are present in the cytosol, nucleus and the extracellular space, and they can bind to many cell types. For galectins present in the extracellular space, it is not clear whether they function as suggested by using recombinant protein added to single cell types. It is thus critical to demonstrate the function of endogenous galectins*.

Galectins are secreted by atypical secretory pathway from different cell types. Therefore, a number of cell types could serve as a source for the secreted galectin-8 that induces RANKL expression by osteoblasts. In experiments aimed to address this question we have now shown that silencing of galectin-8 expression in osteoblasts partially inhibits the expression of RANKL, suggesting that galectin-8, secreted from these cells, could act in an autocrine fashion to induce RANKL expression, along with other stimuli that promote expression of RANKL (new Figure 1). Furthermore, we could show that galectin-8 is present on the surface of bone marrow cells, turning these cells as well into a potential source for secreted galectin-8 that affects osteoblasts (new Figure 1). These findings are also addressed in the Discussion (“Galectins are secreted by atypical secretory pathway […] will be required to resolve this complex issue”).

*3) A key question for understanding the results of this work is whether galectin-8 is expressed in the bone marrow and whether it is secreted and thus present on the cell surfaces. The authors should determine whether galectin-8 could be eluted by treatment with lactose (and not control saccharides)*.

This point was addressed using FACS analysis, with the results added as new Figure 1. We found that galectin-8 is indeed expressed on the surface of bone marrow cells, and can be partially displaced from their surface by TDG, but not by sucrose. We further addressed this point in the first paragraphs of the Results and Discussion sections. This issue was also addressed in the previous paragraph.

*4) If the answer to question #3 is negative, the question is whether the described effects of galectin-8 can in fact occur naturally. If so, the authors need to demonstrate the function of endogenous galectin-8 by using relevant cells with galectin-8 expression knocked down or using cells from galectin-8 knockout mice. The use of transgenic mice overexpressing galectin-8 might be problematic, as it represents an artificial situation with the amounts of galectin-8 released into the environment that might not be reached in wild-type mice*.

The answer to question #3 was positive, nonetheless, we added the results of experiments done with cells where the expression of galectin-8 was silenced by siRNAs (new Figure 1) as mentioned above (question #2).